# EUS-Guided Vascular Interventions

**DOI:** 10.3390/jcm12062165

**Published:** 2023-03-10

**Authors:** Michelle Baliss, Devan Patel, Mahmoud Y. Madi, Ahmad Najdat Bazarbashi

**Affiliations:** 1Division of Gastroenterology, Saint Louis University Hospital, St. Louis, MO 63104, USA; 2Division of Gastroenterology, Washington University in St. Louis, St. Louis, MO 63130, USA

**Keywords:** endoscopic ultrasound, vascular, gastric varices, portal pressure gradient, pseudoaneurysms, portal venous sampling

## Abstract

Endoscopic ultrasound (EUS) has numerous advanced applications as a diagnostic and therapeutic modality in contemporary medicine. Through intraluminal placement, EUS offers a real-time Doppler-guided endoscopic visualization and access to intra-abdominal vasculature, which were previously inaccessible using historical methods. We aim to provide a comprehensive review of key studies on both current and future EUS-guided vascular applications. This review details EUS-based vascular diagnostic techniques of portal pressure measurements in the prognostication of liver disease and portal venous sampling for obtaining circulating tumor cells in the diagnosis of cancer. From an interventional perspective, we describe effective EUS-guided treatments via coiling and cyanoacrylate injections of gastric varices and visceral artery pseudoaneurysms. Specific attention is given to clinical studies on efficacy and procedural techniques described by investigators for each EUS-based application. We explore novel and future emerging EUS-based interventions, such as liver tumor ablation and intrahepatic portosystemic shunt placement.

## 1. Introduction

Since the introduction of endoscopic ultrasound (EUS) as a diagnostic modality in the 1980s, advances in EUS over the years have expanded its applications to an interventional platform by adopting conventional radiological and minimally invasive surgical techniques [1]. While diagnostic EUS interventions have been premised on solid and cystic non-vascular pathology in the foregut, their diagnostic and therapeutic repertoire have recently expanded to vascular pathology. The proximity of the gastrointestinal (GI) tract to major blood vessels in the mediastinum and abdomen and the capability of EUS to provide a real-time Doppler-guided endoscopic visualization of extraluminal structures make EUS uniquely suited for guiding vascular access and therapeutic maneuvers [2]. From a vascular standpoint, EUS guidance can be used to understand vascular anatomy, to determine the presence or absence of vascular flow through a Doppler and waveform analysis, and to intervene with precision at targeted vascular sites that may be less accessible using conventional methods. The ability to visualize and access the portal vein with EUS guidance allows for direct portal pressure measurements, portal venous sampling, and the ablation of liver pathology. EUS-guided vascular coiling offers a minimally invasive alternative to interventional radiology techniques for the management of gastric varices (GVs) and visceral pseudoaneurysmal bleeding. With a shift towards less invasive approaches to the diagnosis and management of GI pathology, the applications of EUS-guided vascular interventions will continue to evolve. Here, we provide a comprehensive review of these promising diagnostic and therapeutic modalities and shed light on possible future applications, including EUS-guided intrahepatic portosystemic shunt placement and EUS-guided cardiopulmonary interventions (Table 1).

## 2. EUS-Guided Vascular Interventions

### 2.1. Diagnostic Applications

#### 2.1.1. EUS-Guided Portal Pressure Measurement

Portal hypertension (PH), most commonly seen as a consequence of cirrhosis, results from complex intrahepatic and extrahepatic pathophysiological alterations that cause an increase in intrahepatic vascular resistance [3]. Identifying the presence and severity of PH in cirrhosis has become an important clinical prognostic tool that can be used to guide management [4]. For example, more severe portal hypertension predicts the presence and risk of bleeding from esophagogastric varices. Currently, the standard method of evaluating clinically significant PH consists of measuring the hepatic venous pressure gradient (HVPG), performed by interventional radiologists. HVPG serves as a surrogate for portal venous pressure (PVP) but is not a direct measurement of the portal pressure gradient (PPG). It is measured by inserting a catheter percutaneously into the hepatic vein and calculating the difference between the free hepatic vein pressure and the wedged hepatic vein pressure (WHVP). In addition to being an indirect measurement of PVP, HVPG is invasive, requires radiation exposure and the use of intravenous contrast, and has been shown to poorly correlate with directly measured portal pressures in patients with non-cirrhotic and presinusoidal causes of PH [5].

The largest barrier to a direct portal pressure measurement is its limited accessibility. Historically, a portal pressure measurement was performed through direct surgical access into the portal vein, which was considered invasive. With the advancement of EUS, we have almost turned full circle, returning our focus on direct portal access for pressure measurements at a less invasive cost. An EUS-guided portal pressure gradient (EUS-PPG) measurement is an alternative novel method of directly measuring the PPG by taking advantage of the proximity of the portal vein to the tip of the echoendoscope in the stomach (Figure 1). With the patient in the supine position under general or monitored anesthesia care, the middle hepatic vein waveform is identified using Doppler flow. A transgastric transhepatic approach is used to introduce a heparin-flushed 25G FNA needle into the hepatic vein. This needle is attached to a manometer, which provides a real-time pressure measurement. A total of three separate hepatic vein pressure recordings are documented, and the average of the three is recorded as the mean hepatic venous pressure (HVP). The FNA needle is then withdrawn, and the same process is repeated with the umbilical portion of the left portal vein, which can be easily identified on EUS and confirmed with a Doppler and waveform analysis (Figure 1). To calculate the PPG, the mean PVP is subtracted from the mean HVP. The concept of a gradient eliminates the potential error associated with using an external zero reference point and with the false elevations in PVP or WHVP caused by factors such as ascites and increased intra-abdominal pressure [6]. While still novel and in its early phases, various animal and human pilot studies conducted over the years have demonstrated the safety and feasibility of EUS-PPG measurements with a high degree of technical success and correlation with HVPG [7,8,9,10]. In a prospective study by Zhang et al., the feasibility and safety of EUS-PPG and the consistency between EUS-PPG and HVPG were explored in 12 patients. EUS-PPG measurements were technically successful in 91.7% of patients, with a high degree of safety and accuracy [7]. The current literature demonstrates high technical success rates. In the largest study of 83 patients, Choi et al. reported a 100% technical success rate. In a more recent study from Zhang et al., the technical success rate was 92%. In both studies, no early- or late-onset adverse events were reported. While these studies reported high technical success rates without significant adverse events, we would like to highlight the potential risks and adverse events of this technique, which include but are not limited to bleeding from the needle puncture (intrahepatic or extrahepatic), bile leak, infection, and peritonitis [7,11]. While EUS-PPG measurements remain similarly invasive to HVPG, the possibility of using EUS as a one-stop shop for PPG measurements, liver biopsy, elastography, and variceal assessment during the same procedure is an attractive option that may ultimately emerge as the standard approach for select patients. Future studies are needed to fully evaluate this modality and compare its outcomes and clinical significance to the current gold-standard HVPG and to non-invasive testing for portal hypertension.

#### 2.1.2. EUS-Guided Portal Venous Sampling

Pancreatic cancer remains one of the most aggressive GI malignancies with a poor prognosis due to a lack of early symptoms and disease biomarkers. The criteria for curative surgical resection in patients with pancreatic cancer are in part dependent on the radiological evaluation of metastatic disease. While the currently available imaging modalities can provide information on the macroscopic evidence of metastasis, they are limited in terms of their ability to identify early micrometastatic disease. This, in turn, could affect the adequacy of prognostication and the prediction of postoperative recurrence risk.

Circulating tumor cells (CTCs) are cells that shed from primary tumors and travel through the systemic circulation to secondary sites where they deposit and act as early seeds for distant metastasis. There is increasing evidence to support the role of CTCs in the early diagnosis of pancreatic cancer and in predicting the risk of metastatic disease [12]. The acquisition of CTCs from vessels proximal to the primary tumor can increase the possibility of detecting enough CTCs to predict the risk of metastatic disease. In the case of pancreatic cancer, which commonly metastasizes to the liver, pancreatic venous drainage into the portal circulation makes the portal vein a potential target for CTC detection.

The utility of CTC acquisition from the mesenteric and portal circulation compared to peripheral blood for prognostication and guidance on the use of adjuvant chemotherapy was initially explored in the surgical setting. In a 2012 study of patients undergoing the surgical resection of colorectal cancer, CTCs were found at a higher rate and count in the mesenteric circulation compared to peripheral blood, and the presence of CTCs was associated with a higher rate of liver metastases at the 3-year follow-up interval [13]. Subsequently, in 2016, the intraoperative acquisition of portal venous blood for CTC enumeration was explored in patients with periampullary and pancreatic adenocarcinoma, and it similarly demonstrated a higher CTC count and detection rate than peripheral venous sampling and a higher rate of liver metastases at the 6-month follow up interval in CTC-positive patients [14].

The surgical collection of CTCs is limited by infrequent patient eligibility for surgery and is prone to inaccuracy due to the potential release of CTCs from intraoperative pancreatic manipulation. Additionally, intraoperative access to the portal vein to collect CTCs in many of these patients is invasive. EUS provides a minimally invasive approach to access the portal vein with precision and to isolate CTCs for risk stratification preoperatively. The portal vein can easily be seen and accessed by a needle from a transduodenal view. In a 2015 single-center cohort study, Catenacci et al. demonstrated the safety and feasibility of EUS-guided portal venous sampling of isolated CTCs in patients with pancreaticobiliary malignancies with a higher yield than peripheral blood samples [15]. These findings were further supported by a prospective study of 40 patients with suspected pancreaticobiliary cancer conducted by Zhang et al. in 2021 [16].

EUS-guided portal venous sampling should be preceded by standard EUS staging and diagnostic confirmation with EUS-FNA (Figure 2). Cross-sectional imaging should be studied to evaluate for aberrant anatomy or possible contraindications to needle access. For blood sample acquisition, Chapman et al. recommend the use of a 19G EUS-FNA needle for improved blood flow, which reduces clotting and the time spent within the vessel. Before introducing the needle into the portal vein, a Doppler-guided assessment of vessel anatomy and a confirmation of vessel patency and flow should be performed. Special care should be taken to avoid needle contact with any metastatic lesions or lymph nodes. Negative suction should be used during aspiration once the portal vein is accessed. Once the sample is acquired, the needle is slowly withdrawn with close attention to the intrahepatic needle track and puncture site using Doppler visualization to identify sites at high risk of persistent bleeding.

### 2.2. Therapeutic Applications

#### 2.2.1. EUS-Guided Gastric Variceal Coiling

Gastroesophageal varices (GVs) are dilated portosystemic collateral veins that can cause significant gastrointestinal bleeding in patients with cirrhosis and portal hypertension. Although GVs represent 20% of variceal bleeding, they are associated with poorer outcomes, including more severe bleeding at index presentation, higher transfusion requirements, and an increased risk of rebleeding compared to esophageal varices (EVs) [17,18]. Despite having worse outcomes, there exist sparse evidenced-based guidelines for the management of GVs (actively bleeding and prophylactic GVs), especially when compared to EV management. 

While various GV classification systems exist, the Sarin classification has been the most used, particularly when it comes to management decisions (Figure 3) [19]. GOV-1 is treated similarly to EV, such as with endoscopic band ligation. Meanwhile, GOV-2, IGV-2, and IGV-2 can be treated with direct endoscopic injection therapies, transjugular intrahepatic portosystemic shunts (TIPSs), or balloon retrograde transvenous obliteration (BRTO). However, these treatments have significant limitations, such as recurrent bleeding, systemic embolization, limited feasibility in GVs associated with splenic vein thrombosis, and occasionally limited resources to these modalities [18]. More recently, EUS-guided GV management has emerged as an alternative intervention, with promising clinical success and low risks of complications and recurrent bleeding [20,21,22,23,24].

EUS can assist in the identification of GVs and reveal important characteristics, such as varix size and flow, which can provide information to guide optimal management at the point of care [17]. Flow information is especially useful for EUS-guided cyanoacrylate (CYA) glue injections to ensure that an optimal amount of CYA is delivered for obturation and the risk reduction of CYA-related embolization. EUS GV coiling was later introduced in 2010 as a promising therapy and has recently been adopted as the more common EUS-guided technique for GV management, including for actively bleeding GVs and for prophylaxis [17]. EUS-guided coil therapy may include the deployment of coils alone or coils alongside injectate, such as acrylate polymers (cyanoacrylate) or an absorbable gelatin sponge [25]. 

While no standardized technique for EUS-guided coiling exists, many institutions have adopted common steps to ensure a safe deployment, high clinical success, and a minimal risk of adverse events. Patients are in the left lateral position and often sedated with general anesthesia. An upper endoscopy is performed to evaluate the location and size of the gastric varices and to obtain information on concurrent esophageal varices. Antibiotics are often administered for prophylaxis. Next, a linear echoendoscope is advanced to the distal esophagus or gastric fundus to assess the anatomy of the GV and feeding vessels along with flow patterns. Water is infused in the fundus to assist with the better delineation of the GV to enhance acoustics and improve ultrasound image quality. Coils of various lengths and diameters are delivered through a 19G or 22G needle into the varix and/or feeder vessel under EUS guidance. The coils are advanced into the varix with the assistance of a stylet, under endosonographic and sometimes fluoroscopic guidance. The number of coils deployed is often operator-dependent and relies on evidence of a diminished or abrupt cessation of Doppler flow. An iodinated contrast agent can be injected into varices after coil deployment to ensure that there is no evidence of a persistent shunt. Cyanoacrylate can then be injected as adjunctive therapy [17,18,25] (Figure 4).

While cyanoacrylate has been proven to be an effective therapy for the treatment of GVs, with or without coil use, it does carry certain limitations, such as the risk of damaging endoscopes and causing adverse events, including rebleeding and systemic embolization. Additionally, cyanoacrylate can polymerize early and lead to the deroofing of the varix when the needle is pulled back. Lipiodol can assist in preventing early polymerization. More recently, an absorbable gelatin sponge has been used as an alternative to cyanoacrylate, as it does not carry similar risks. Bazarbashi et al. recently evaluated the use of absorbable gelatin sponges (such as Gelfoam or Surgiflo) as adjunctive therapy with coils (instead of cyanoacrylate). An absorbable gelatin sponge has been used for intravascular thrombolysis with IR and carries low risks of embolization. In their matched cohort study, Bazarbashi et al. demonstrated the superiority of AGS to cyanoacrylate for the treatment of GVs [25].

Surveillance EUS to monitor GVs after coil therapy is typically carried out at 1, 6, and 12 months. Repeat EUS-guided coil and gel injections may be needed depending on the response after index endoscopy and the size of the varices and ongoing Doppler flow. The complications of EUS-guided coils and therapy include the systemic gel embolization of concurrent injectate (cyanoacrylate embolization), transient abdominal pain, minor bleeding from the needle site puncture, and benign coil tip extrusion [17,25]. To date, based on the literature and to the best of our knowledge, there is no evidence of coil migration after EUS-guided coil therapy for GVs.

EUS-guided coil therapy is limited by the availability of expertise and EUS equipment and a lack of evidence on coil size and the requisite number for optimal outcomes. Despite these limitations, EUS-guided coiling has been demonstrated in multiple studies to obturate gastric varices with excellent outcomes, including low rates of rebleeding and adverse events [17,18].

#### 2.2.2. EUS-Guided Arterial Pseudoaneurysm Coiling

Visceral arterial pseudoaneurysms (VAPAs) are rare, abnormally dilated arteries associated with significant morbidity and mortality. Intra-abdominal organ pathologies, such as surgery and pancreatitis, can lead to the development of VAPAs. The commonly involved arteries include the splenic, hepatic, superior mesenteric, and pancreaticoduodenal arteries. Unlike true aneurysms, VAPAs represent ballooned blood vessels with thin walls, resulting in a higher risk of rupture and significant bleeding. In chronic pancreatitis, studies demonstrate a risk of rupture up to 50% and a mortality post-rupture between 15 and 40% [25].

Interventional radiology procedures and surgery have been historically utilized to treat these lesions. However, these procedures can be technically challenging, especially in cases of small pseudoaneurysms not detected by imaging and in anatomically difficult locations in which endovascular methods may not be feasible. EUS may overcome such barriers by providing an improved visualization and access to previously inaccessible abdominal pseudoaneurysmal lesions. In turn, VAPAs can be directly injected with EUS-guided devices, such as coils, thrombin, and glue, in a minimally invasive manner, resulting in an effective and safe therapy [26].

EUS-guided pseudoaneurysm coiling follows a similar technique to that of EUS-guided coil therapy for GVs (Figure 5). An echoendoscope is introduced into the stomach. The Doppler technique is used to detect the VAPA, including a waveform analysis, and to accurately measure the pseudoaneurysm to guide coil placement (the diameter of the coil and the number of coils). A 19G fine-needle aspiration needle is inserted directly into the VAPA. Once secure, the needle stylet is removed, and embolization coils are loaded via the FNA needle into the VAPA and can subsequently be injected for further treatment. Further coils are injected until VAPA obliteration occurs, which can be confirmed using Doppler technique.

In 2018, Rai et al. described a standard EUS-guided coiling approach in splenic artery pseudoaneurysm treatment [27]. Coils were deployed under EUS guidance followed by an injection of N-butyl-2-cyanoacrylate glue. All patients achieved both technical and clinical success, as defined by VAPA obliteration on a 12-week follow-up EUS and no evidence of blood loss. Patients required one–two treatment sessions with one–three coils inserted. They reported no procedure-related adverse events or deaths [27]. Comparable results of a high technical success have been reported in EUS-guided thrombin injections and EUS-guided salvage therapy in previously treated splenic artery pseudoaneurysms via an endovascular approach, reflecting the effective nature of the EUS-guided treatment of VAPAs [26,28,29].

However, EUS-guided techniques may be limited by echoendoscopic detection, the possible need for repeat therapies to achieve obliteration, and lesion accessibility [30]. The complications of EUS-guided therapies have been reported to be post-procedural pain; rebleeding, especially in incompletely obliterated VAPAs; coil migration and erosion (as can be seen in IR-guided coil therapy); infection; and thrombosis [26]. Coil erosion can be particularly devastating. After a patient presented with a complicated coil migration into his stomach via a gastrosplenic artery fistula, he underwent a partial gastrectomy, distal pancreatectomy, and splenic artery pseudoaneurysm resection as treatment [31].

EUS-guided coil embolization represents a promising effective application of endoscopic ultrasound in the treatment of highly morbid intra-abdominal vascular pseudoaneurysms that should be added to the armamentarium of VAPA management, particularly when standard IR-guided therapies are limited or contraindicated.

### 2.3. Future Directions

Future applications of EUS-guided vascular interventions are expected to emerge with further advances in endoscopic technology and the availability of longer-term data. Some applications currently being explored, include EUS-guided liver tumor ablation, EUS-guided intrahepatic portosystemic shunt placement, EUS-guided cardiac interventions, and EUS-guided thrombolysis of pulmonary arterial thrombosis [32,33].

#### 2.3.1. EUS-Guided Liver Tumor Ablation

Various innovative ablative techniques are routinely utilized in the treatment of primary and metastatic liver tumors. Whether for cure or for palliation, the percutaneous ablation of liver tumors under ultrasound, computed tomography, or magnetic resonance guidance aims to detrimentally impact a pathologic lesion whilst sparing the surrounding tissues. In certain instances, the application of percutaneous ablative techniques is limited by difficult anatomical locations, such as the caudate and left lobe of the liver. EUS-guided liver tumor ablation, though primarily experimental and in its early stages, is a promising addition to the therapeutic repertoire with an enticing potential for advancement in the coming years. This allows for a safe, effective, and readily available intervention for tumors in difficult locations when alternative methods may not be feasible [34].

EUS-guided liver tumor ablation can be accomplished using different techniques [35] (Figure 6). An EUS fine-needle injection (FNI) entails the injection of sclerosing agents, such as ethanol gels or antitumor agents, directly into the tumor cells or the portal circulation. Ethanol gel, the most commonly used sclerosing agent, not only has a destructive effect on tumor cells but also induces local vasculitis leading to a reduction in recurrence rates. EUS thermal ablation is another option with radiofrequency ablation (RFA), cryotherapy, and interstitial laser coagulation (ILC), where energy is applied directly to liver tumors. With RFA, the goal is to generate and sustain 50–100 °C in the target lesion to achieve adequate ablation. Nd:YAG is the predominantly used type of laser in ILC, with small enough fibers allowing passage into the EUS scope and through FNA needles. Cryotherapy is a technique that primarily damages tissues by freezing followed by thawing. This technique is of futuristic interest, as it has not yet been used to ablate liver tumors. Other techniques that may theoretically be of value include EUS-brachytherapy with radioactive seeds or gel and EUS-guided photodynamic therapy. EUS portal vein embolization allows for access to the left (supplying segments 5, 6, 7, and 8) and right portal veins (supplying segments 2, 3, and 4) given its clear visibility on transduodenal views. From a transgastric view, the umbilical portion of the left portal vein can be accessed. While this will allow for multi-segment embolization, the feasibility for single-liver-segment EUS-guided embolization, by accessing the sub-branches of the left and right portal veins is not known. This would be prudent for oncological planning and may pose a limitation to EUS-guided ablation [36].

Despite the promising nature of EUS-guided liver tumor ablation, the limitations must be acknowledged at this time. These include the current experimental nature of the majority of the methods described in our review and the lack of the current primary role of interventions, such as EUS-FNI, in liver tumor ablation, in addition to the smaller yet present risk of malignant seeding when compared to percutaneous techniques [37].

As our technologies continue to rapidly advance, the production and development of EUS-specific needle ablative systems while maintaining flexibility and limiting the diameter are required. Accurate mapping methods are also in demand to allow for precise therapy. More importantly, further research with comparative studies and randomized controlled clinical trials must be conducted to ensure the effectiveness, safety, and applicability of EUS-guided liver tumor ablation.

#### 2.3.2. EUS-Guided Intrahepatic Portosystemic Shunt Placement

Transjugular intrahepatic portosystemic shunt (TIPS) placement remains the most frequently performed procedure to alleviate portal hypertension (PH) and its consequences, with high technical success and efficacy and a low risk of adverse events compared to surgical shunting techniques [38,39]. PH drives the major complications in cirrhosis leading to increased readmissions and mortality. With TIPS, an angiographic technique involving transjugular access and the advancement of a catheter and a guidewire through the right heart to the inferior vena cava is employed to create an artificial low-resistance channel between the portal and hepatic veins, thereby directing blood flow to the systemic circulation to alleviate PH and reduce its risk of complications.

While TIPS is largely safe and effective, adverse events can occur due to the route of access. These complications, while rare, include inadvertent arterial, tracheal, and biliary injuries and cardiac conduction and rhythm events [40].

Advances in EUS have led to experimental and animal model work evaluating the possible role of an EUS-guided portosystemic shunt (EIPS). While in its infancy with much work to be carried out, EIPS may have a potential role in the future management of patients with PH, particularly when vascular access with TIPS carries a high risk of complications. EIPS does not require entrance into the right heart or inferior vena cava; does not involve radiation exposure; and can be combined with EUS-guided interventions, such as direct portal pressure measurements or GV management in a one-stop shop fashion. Other major technical differences of EIPS from conventional TIPS include transluminal access to the hepatic vein from the upper gastrointestinal tract instead of transvascular catheterization, an EUS-guided puncture instead of a radiologic and percutaneous US-guided puncture, and stent type.

The technique of EIPS was first described by Buscaglia et al. in 2009 in a liver porcine model using a self-expandable tubular metal stent [41]. In this study, the fully expanded stent did not adequately cover the area between the PV and HV in some animals, and a second stent was deployed as a bridge. No major complications were noted, and a 2-week survival period was reported. Subsequently, in 2011, Binmoeller et al. reported a similar EIPS technique for the successful novel transgastric deployment of a lumen-apposing metal stent (LAMS) in a non-survival porcine model [42].

In 2017, Schulman et al. conducted an animal survival study model in which the high technical feasibility of EIPS (with a technical success rate of 100%) combined with a simultaneous direct digital PVP measurement was demonstrated [43]. A lumen-apposing metal stent was also used in this non-cirrhotic animal model study, and while this did confirm its feasibility, this technique remains primitive with many parameters that require further investigation in humans, particularly in those with cirrhosis in whom the risks of coagulopathy and infection are high. Some complications seen in this study included the development of liver abscesses in two of the animals; however, prophylactic antibiotics were not utilized. In-stent thrombosis was also seen in several animals; thus, stent modifications for the purpose of intravascular use may be needed.

#### 2.3.3. EUS-Guided Cardiac Interventions and Thrombolysis

The location of the heart and pulmonary vessels in proximity to the esophagus has prompted the early exploration of EUS-guided transesophageal cardiopulmonary interventions. In 2007, Fritscher-Ravens et al. described the use of EUS in porcine models to guide a puncture of the heart and ultimately performed radiofrequency ablative therapy, pericardial fluid aspiration, cardiac tumor puncture, and pacing wire insertion. No arrhythmias were noted during the procedure, and no cardiac abnormalities resulted [44]. Subsequently, the successful EUS-guided drainage of a pericardial cyst was reported by Larghi et al. in 2009 [45]. In 2019, Romero-Castro et al. reported the use of an EUS-guided biopsy to confirm the diagnosis of a right atrial lymphoma and right atrial myxoma [32]. More recently, an EUS-guided biopsy of an intraventricular fibroadipose mass was reported by Mehta et al. in 2022 [46].

The use of EUS to direct vascular thrombolysis was explored by Sharma et al. in 2017 in the case of an acute portal venous thrombus [47]. EUS was used to guide a puncture into the superior mesenteric vein followed by the placement of a cannula into the vein. The cannula was routed through the nose and used to infuse a thrombolytic agent continuously. Although there was a reported radiological improvement, subsequent bleeding was reported from the site of injection, which was treated by inflating a G-EYE balloon after failure to achieve hemostasis with epinephrine and hemostatic clip placement. Sharma et al. later reported seven cases of EUS-guided thrombolysis for acute portal vein thrombosis in 2019 with a 100% technical success rate [48]. In five cases, the EUS-guided puncture allowed access to the portal venous system, and continuous catheter thrombolysis was administered for 72 h to 10 days via a cannula, which was routed through the nares. In the remaining three cases, bolus injections were administered. For the bolus injections, the splenic vein was accessed through a puncture of the body of the stomach, the portal vein was accessed through a duodenal bulb puncture, and the superior mesenteric vein was accessed through the pancreas. Among the seven cases, one patient experienced catheter site bleeding with the catheter in situ, one patient experienced mild oozing following catheter removal, and one patient developed a splenic infarct on day 7. In 2019, Somani et al. similarly used EUS-guided thrombolysis in a patient with superior mesenteric vein and pulmonary artery thrombosis in whom systemic anticoagulation was contraindicated in light of a recent hemorrhagic stroke. This resulted in a substantial reduction in thrombus size without reported complications [33].

EUS-guided cardiac interventions and vascular thrombolysis are experimental interventions seen in animal model studies and isolated human case reports. While unlikely to be common approaches given the less invasive and more well-established access that currently exists, EUS can provide rare diagnostic and therapeutic benefits for cardiac pathology and for vascular thrombolysis.

### 2.4. Limitations and Complications

While studies have shown promising results for the safety and technical success of EUS-guided vascular interventions, it is important to note that this field remains in its infancy, and much of the data available are limited to case series and retrospective single-center studies. Therefore, it is prudent that we highlight the potential pitfalls and limitations of EUS-guided vascular therapies. First and foremost, EUS-guided vascular interventions, when compared to gold-standard therapies, are compared to IR-guided vascular interventions. IR-guided vascular interventions have proven to be extremely successful, with high clinical and technical success rates. IR-guided interventions allow for safe access to various splanchnic and visceral vessels through the percutaneous route. We envision the role of EUS-guided vascular therapy to be supplemental, rather than a substitute, to IR-guided vascular interventions. Another limitation includes the limited knowledge on the vascular anatomy of the GI tract when applied to EUS. This is an evolving field, but much work is needed to better delineate the vascular anatomy of the GI tract to ensure safe and effective access and subsequent therapies. The third limitation is that many of the tools and techniques available for EUS are not specifically designed for vascular interventions and that many of the tools used for EUS vascular access and treatment are adopted from those used in interventional radiology (for example, GV coiling).

In terms of complications, we want to highlight that, while these are rare, they can be significant. The risks and complications that need to be highlighted include bleeding (from the target vessel or puncture site); infection, including peritonitis and liver abscess formation; systemic embolization; and visceral perforation.

## 3. Conclusions

The applications of EUS-guided vascular interventions continue to evolve, affording multiple therapeutic avenues for various conditions. The unique location of the GI tract in proximity to major vascular structures allows for the use of EUS to guide these vascular interventions and offers potential alternatives to standard diagnostic and treatment modalities performed by interventional radiology. Although smaller-scale studies have shown promising safety results, clinical efficacy, and technical success rates, future larger-scale studies are needed to demonstrate how these parameters compare with the currently available approaches for the management of these conditions. A better understanding of vascular anatomy, improved EUS resolution and acoustics with a vascular assisted analysis, and the development of vascular-friendly EUS-deployed stents and coils will hopefully assist with the advancement of this promising field.

## Figures and Tables

**Figure 1 jcm-12-02165-f001:**
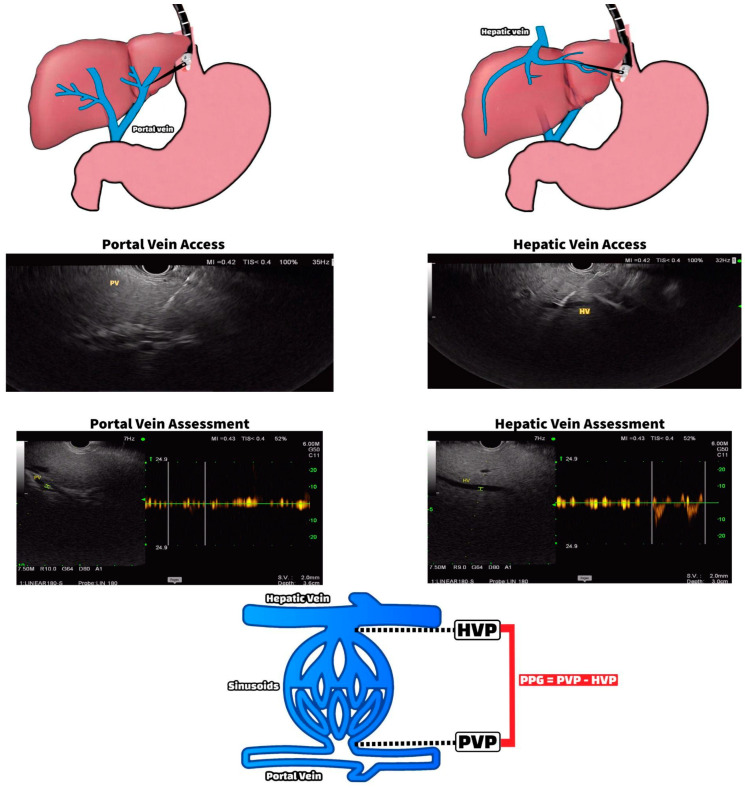
EUS-guided portal pressure gradient measurement.

**Figure 2 jcm-12-02165-f002:**
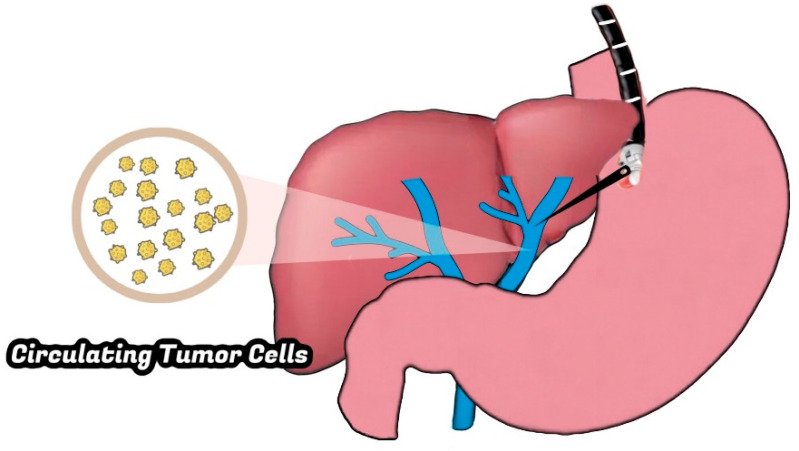
EUS-guided portal venous sampling for circulating tumor cells.

**Figure 3 jcm-12-02165-f003:**
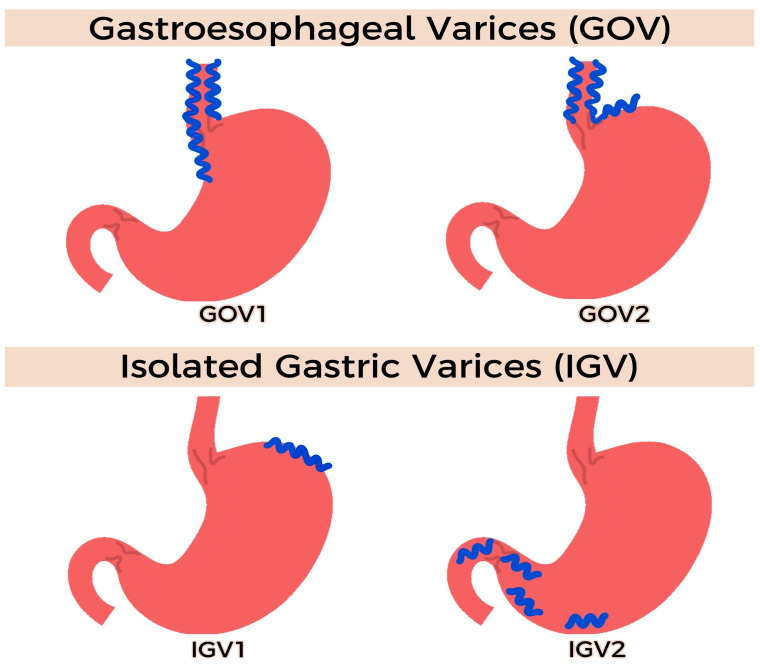
Sarin classification of gastric varices.

**Figure 4 jcm-12-02165-f004:**
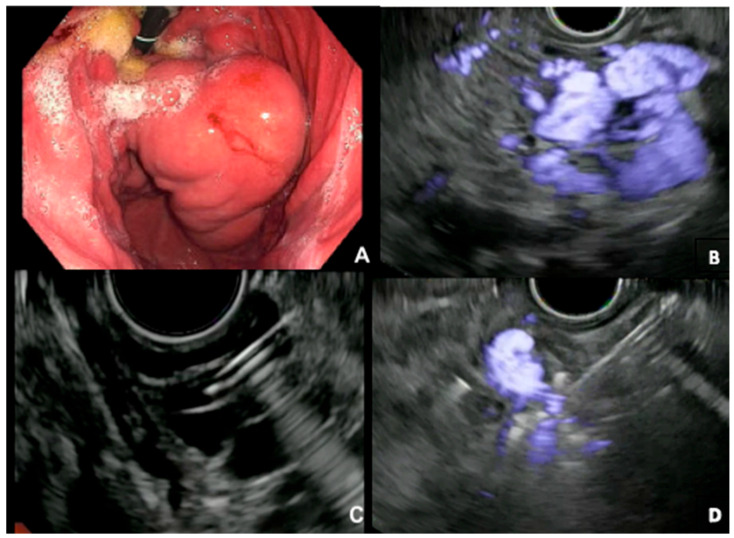
(**A**) Endoscopic examination of large GV on retroflexion. (**B**) EUS confirming Doppler flow within large varices. (**C**) Needle access into GV under endosonographic guidance with coil deployment. (**D**) Diminished Doppler flow on EUS after coil injection.

**Figure 5 jcm-12-02165-f005:**
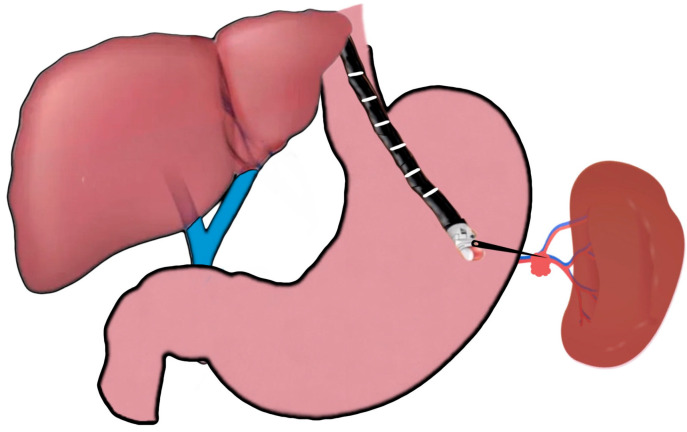
EUS-guided splenic artery embolization.

**Figure 6 jcm-12-02165-f006:**
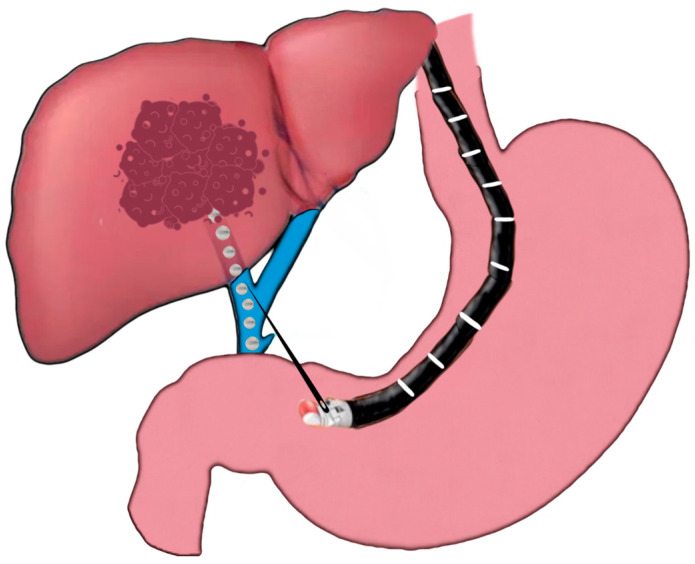
EUS-guided liver tumor ablation.

**Table 1 jcm-12-02165-t001:** General overview of current and future EUS-guided diagnostic and therapeutic techniques.

EUS-Guided Vascular Interventions	
Category	Intervention
Diagnostic	Portal pressure measurement
	Portal venous sampling
Therapeutic	Gastric variceal coiling
	Arterial pseudoaneurysm coiling
Future directions	Liver tumor ablation
	Intrahepatic portosystemic shunt placement

## Data Availability

Research data available from publicly archived datasets available online.

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
