# Peer review of "EUS-Guided Vascular Interventions"

_jcm, 2023, doi:10.3390/jcm12062165_

Round 1

Reviewer 1 Report

Dear Editor

I read with interest the submitted manuscript " EUS-Guided Vascular Interventions" submitted to the JCM. I suggest accepting but with minor edits. My comments are:

1- While the article summarises the current diagnostic and therapeutic uses of EUS, there is no mention of the possible complications and risks associated with EUS vascular intervention. Can the author cite the possible complications and use literature to provide evidence?

2- I suggest adding a paragraph about the future use of EUS in direct thrombolysis of acute portal vein thrombosis. this is an emerging use of EUS in endoscopy and is potentially life-saving. Use the following references:

  • Sharma, Malay MBBS, MD, DM; Somani, Piyush MBBS, MD, DM; Jindal, Saurabh MBBS, MD, DM. EUS-Guided Continuous Catheter Thrombolysis of Portal Venous System: 1614. American Journal of Gastroenterology 112():p S877,S879, October 2017.

  • Benmassaoud A, AlRubaiy L, Yu D, Chowdary P, Sekhar M, Parikh P, Finkel J, See TC, O'Beirne J, Leithead JA, Patch D. A stepwise thrombolysis regimen in the management of acute portal vein thrombosis in patients with evidence of intestinal ischaemia. Aliment Pharmacol Ther. 2019 Nov;50(9):1049-1058. doi: 10.1111/apt.15479.

Best regards

Author Response

Response to Reviewer 1 Comments

Point 1: While the article summarizes the current diagnostic and therapeutic uses of EUS, there is no mention of the possible complications and risks associated with EUS vascular intervention. Can the author cite the possible complications and use literature to provide evidence?

Response 1: Thank you for your input. We have added a final section titled “limitations and complications” that addresses some of the reported complications of EUS-guided vascular interventions. 

Point 2: I suggest adding a paragraph about the future use of EUS in direct thrombolysis of acute portal vein thrombosis. This is an emerging use of EUS in endoscopy and is potentially life-saving. 

Response 2:  Thank you for this suggestion. EUS guided thrombolysis was mentioned briefly in the original submission under section 2.6. We elaborated on the topic using one of the references which were kindly provided (Sharma et al). 

Reviewer 2 Report

This review by Baliss et al provides insights into the EUS-guided vascular interventions. Although there are a number of articles delineating this work, things which could add value is focusing on the vascular component. As noted, the pictorial representations are of value in depicting the needle sizes and access routes. 

Introduction: well written

EUS-guided PPM: Please add technical and clinical success rates. Also provide the complications noted with this access. An important note to add is NOT only the advantages but the adverse events with these procedures which has to be taken to provide a holistic use in the future. 

GOV: There is a lot of data on this and hence no further details need to be added. 

Liver tumor ablation: Please add info on segmental targeting and if any limitation via EUS for access lesions towards the dome. 

At the end (prior to the conclusion): Please add limitations section

Author Response

Response to Reviewer 2 Comments: 

Point 1: EUS-guided PPM: Please add technical and clinical success rates. Also provide the complications noted with this access. An important note to add is NOT only the advantages but the adverse events with these procedures which has to be taken to provide a holistic use in the future. 

Response 1: Thank you. We have added a paragraph about this.

Point 2: GOV: There is a lot of data on this and hence no further details need to be added. 

Response 2: Thank you.

Point 3: Liver tumor ablation: Please add info on segmental targeting and if any limitation via EUS for access lesions towards the dome. 

Response 3: Thank you. We have added a paragraph commenting on segmental targeting.

Point 4: At the end (prior to the conclusion): Please add limitations section

Response 4: Thank you for your input. We have added a final section titled “limitations and complications” that addresses some of the reported complications of EUS-guided vascular interventions.

Reviewer 3 Report

The paper is well written and fine to accept with possible minor revision.

Possible edits are

  1. p. 7, line 205:  “A 19G or 22G needle, back loaded with coils of various lengths and diameters” should be changed to “Coils of various lengths and diameters are delivered through a  19G or 22G needle since some operators do not backload
  2. p 7, line 211: “Cyanoacrylate or absorbable gelatin  sponge can then be injected as adjunctive therapy”  Delete  “or absorbable gelatin  sponge” and communicate authors’ experience with sponge    separately and need for further validation
  3. p8: consider replacing fig 4D with endoscopic view after obliteration
  4. p 9 ‘Future Directions’ misplaced - put at end?

Author Response

Response to Reviewer  3 Comments

Point 1: p. 7, line 205:  “A 19G or 22G needle, back loaded with coils of various lengths and diameters” should be changed to “Coils of various lengths and diameters are delivered through a  19G or 22G needle since some operators do not backload

Response 1: The recommended edit has been made, and the sentence now reads: “Coils of various lengths and diameters are delivered through a  19G or 22G needle”.

Point 2: p 7, line 211: “Cyanoacrylate or absorbable gelatin  sponge can then be injected as adjunctive therapy”  Delete  “or absorbable gelatin  sponge” and communicate authors’ experience with sponge  separately and need for further validation

Response 2: “Absorbable gelatin sponge” has been deleted. A separate paragraph was added to discuss experience and literature with absorbable gelatin sponge.

Point 3: consider replacing fig 4D with endoscopic view after obliteration

Response 3:  Unfortunately, we do not have endoscopic images after coil embolization for this particular series of images.

Point 4: p 9 ‘Future Directions’ misplaced - put at end?

Response 4: The “Future Directions” section is the final section detailing EUS-guided vascular interventions under heading “C”. The subheadings that follow further elaborate on applications of EUS that fall under the broad category of “Future directions”.

Reviewer 4 Report

Thank you for an extensive review of the EUS-guided vascular interventions in such a well-structured manuscript. However, there are few points that should be mentioned.

1. In the EUS-guided portal venous sampling section, does the technique performed through the liver or through the peritoneum as shown in the figure? In the transperitoneal approach, is there any risk for bleeding?

2. In the EUS-guided coiling for gastric varices treatment, is there any tips or technique to avoid systemic coil embolization? How to In cases that cyanoacrylate injection is performed, is there any risk of endoscope injury or needle occlusion?

Author Response

Response to Reviewer 4 Comments

Point 1: In the EUS-guided portal venous sampling section, does the technique performed through the liver or through the peritoneum as shown in the figure? In the transperitoneal approach, is there any risk for bleeding?

Response 1: Thank you for this question. We apologize if the image appeared to show that this was a trans-peritoneal stick. We have amended this to show that this technique is transparenchymal, unusually into the umbilical portion of the left portal vein.

Point 2: In the EUS-guided coiling for gastric varices treatment, is there any tips or technique to avoid systemic coil embolization? How to In cases that cyanoacrylate injection is performed, is there any risk of endoscope injury or needle occlusion?

Response 2: Thank you for bringing this great point up. To date, based on literature and as far as we know, there is no evidence of coil migration post EUS guided coil therapy for GV. There is data on cyanoacrylate embolization however no data on coils migrating. For cyanoacrylate embolization, many believe that the use of coils reduce the risk of migration, as these coils act as a scaffold preventing cyanoacrylate from migrating.  Other limitations of cyanoacrylate is that it can be damaging to endoscopes if used incorrectly. Additionally, cyanoacrylate can polymerize early and lead to deroofing of the varix when the needle is pulled back. Lipiodol can assist in preventing early polymerization. More recently, absorbable gelatin sponge has been used as an alternative to cyanoacrylate as it does not carry similar risks. We have added a paragraph about this in the edits.